# Oil and Flower Production in *Rosa damascena trigintipetala Dieck* under Salinity Stress in Taif Region, Saudi Arabia

Mohamed E. El-Sharnouby [1,*], Metwally M. Montaser [2] and Sliai M. Abdallah [3]

1 Department of Biotechnology, College of Science, Taif University, P.O. Box 11099, Taif 21944, Saudi Arabia
2 Science and Technology Department, University College of Ranyah, Taif University, Ranyah 21975, Saudi Arabia; m.montaser@tu.edu.sa
3 Department of Biology, Turabah University College, Taif University, Taif 21944, Saudi Arabia; a.sliai@tu.edu.sa
* Correspondence: m.sharnouby@tu.edu.sa

**Abstract:** The flower industry depends on oil and fragrance, which is addressed in the current work. Different concentrations of NaCl (0, 250, 500, 1000, and 1500 ppm) were applied to Taif rose plants (*Rosa damascena var. trigintipetala Dieck*) to evaluate their effects on growth and essential oil content. Results clearly indicated the highest survival percentage (98.3%) was seen in untreated plants compared to plants under salinity stress. Moreover, increasing the NaCl levels induced an adverse effect on the growth parameters of Taif rose plants, while some essential oil contents were increased to the maximum degree of their tolerance to salinity stress. The extracted essential oils were analyzed using GC/MS. The essential oils of Taif rose plants treated with 500 ppm NaCl recorded the highest values of citronellol, geraniol and phenylethyl alcohol contents (16.56, 8.67 and 9.87%), respectively. NaCl at 250 ppm produced the highest values of heneicosane (13.12%), and then decreased to the lowest value (7.79%) with the increase of NaCl to 1500 NaCl, compared to the control and other NaCl levels. The current results could highlight the impact of salinity stress on *Rosa damascena Miller var. trigintipetala Dieck* for better economic and industrial applications.

**Keywords:** citronellol; geraniol; NaCl; survival; Taif rose plants

## 1. Introduction

Globally, salinization is a major challenge to sustainable agricultural production and food security. Mismanagement of agricultural lands and overexploitation of water resources in arid climates have been highlighted as some of the major causes of soil salinization [1]. In addition to reducing net arable lands, soil salinization has serious implications for food security and the livelihoods of farmers, potentially impacting the regional and national economy [2]. Salinity is one of the main limiting factors for agricultural and natural production. Plants exposed to salinity at high levels induce more biochemical and physiological changes, leading to modifications in the function of cell membranes [3]. High salt content in irrigating water causes a reduction in plant growth and crop yields [4,5].

Salinity stress because of excessive Cl− and Na+ ions reduce plant growth, and has an adverse effect on plant physiology [6]. There are many plants tolerant to NaCl that can survive and adapt to high salinity, and induce physiological and biochemical changes [7]. Zhang et al. [8] studied the physiological responses of *Populus* sp. plants to different salinity levels. Chen et al. reported the use of MS/MS technology in the evaluation of salt-stress tolerance on *Apocynum venetum* L. plants for Chinese medicine, to get a total of 43 bioactive contents in response to salinity [9], while others recently reported the salinity impact on fatty acid composition in different aromatic plants, such as coriander, sweet marjoram, and sweet basil [10–12].

The majority of recent research on salt tolerance in plants focuses on determining the genes involved in the molecular mechanisms of tolerance, and some use transgenic plants in order to get a better response to salinity [13].

Recently, many studies were conducted to detect the effects of abiotic factors on the production of essential oil contents in aromatic species [14]. Lui et al. [15] used GC-MS analysis and compared the phytochemical characteristics of both *Rosa roxburghii* and *R. sterilis*. They reported 91 different components between *R. roxburghii* and *R. sterilis* fruit [15]. Others reported the major essential oil content to be nonadecane (15%), citronellol (13%), heneicosane (17%), and geraniol (9%). However, the main compounds of the essential oils were nonadecene (4–4.55%), β-citronellol (30–31%), trans-geraniol (20–21%), n-heneicosane (8–9%), and phenylethyl alcohol (4–4.16%) [16]. Ryu's group investigated the volatile compositions of 12 rose (*Rosa hybrida*) flower-color mutant variants and their original cultivars [17]. It is worth stating that plants grown under open sun with a black polyethylene mulch produced a good quality of damask rose [18].

Roses are one of the most important floricultural crops, and their essential oils are used for cosmetics and aromatherapy. *Rosa damascena Miller var. trigintipetala Dieck* is considered one of the best species for producing rose oil. It is commonly known as the damask rose, and is considered the most important commercial crop in the Taif region [11]. Saudi Arabia is flourishing in the Taif region through flower cultivation [12]. Kürkçüoglu et al. (2013) reported that the fresh flowers of *Rosa damascena Miller var. trigintipetala Dieck* cultivated in the Taif region are the source of Taif rose oil [19]. Taif roses have been processed into attar of roses and rose water for the past two centuries. Taif rose oil and rose water have become important and commercially valuable products, and the oil can also help in the treatment of depression [20].

The quality of the essential oil of the damask rose is due to the high percentage of monoterpene alcohols such as citronellol, phenyl ethyl alcohol, hydrocarbons, and nonadecene. The growth and yield of damask roses are affected by many agronomic factors [21].

The present work was designed to evaluate the tolerance of Taif rose plants to different concentrations of NaCl, considering plant growth and the essential oil content of Taif rose plants.

## 2. Materials and Methods

Experiments were carried out at the biotechnology department of the Faculty of Science, Taif University, during 2020. The study investigated the growth of Taif rose plants (*Rosa damascena trigintipetala Dieck*) cultured in the Taif region for 6 months at different salinity treatment levels. Rose plant leaves were taken to determine growth parameters regarding the number of leaves, number of flowers, height of plants, survival percentage, and essential oil content.

### 2.1. Experiment Design

Nodes of the Taif rose plant (*Rosa damascena trigintipetala Dieck*) were cultured in plastic pots (10 cm height, 82 cm top circumference, and 70 cm bottom circumference), containing 2.5 kg of a mix of sand and peat moss (1:1) (ARBER® Horticulture, Italy). The plants were cultured at five salinity levels (0, 250, 500, 1000, and 1500 ppm NaCl). Fifty plants were randomly divided into 5 groups of 10 pots each (1 plant/pot) according to Attia et al. [22], with modifications. Each treatment was applied to a group of plants (10 plants/group). The plants were irrigated once (300 mL tape water per pot) and then treated with saline water every three days. Cultures were kept in an open field during the summer of 2019 at Taif City, KSA.

### 2.2. Measurement of Growth Parameters

After 6 months of Taif rose plant (*Rosa damascena trigintipetala Dieck*) cultivation, stress tolerance was evaluated based on morphological parameters such as plant height (cm),

number of leaves (leaf/plant) and flowers (rose/plant), and plant survival (live plant/total 250 plants) was recorded.

### 2.3. Extraction Method of Essential Oil

Twenty mg of rose essential oil was extracted from 100 gm flower samples, and dissolved in 1 mL N- hexane for 4 h. Triplicates were used for each sample. Essential oil samples used for the GC/MS analysis were dissolved in dichloromethane according to Xiao et al. [23].

### 2.4. Gas Chromatography of Taif Rose Essential Oil

The analysis of the samples was performed using a Varian GC-MS system (Model Varian CP 3800, Varian Saturn 2200) at 70e V, 250 °C, using helium as the carrier gas (flow rate of 1 mL/min). The oven temperature was programmed for 5 min at 260 °C. Identification of components as concrete and absolute oils was based on matching with Wiley and the NIST electronic library.

### 2.5. Statistical Analysis

The significant difference between factors was ascertained. Leven's test was used to determine data homogeneity, followed by the least significant difference with Post-Hoc one-way ANOVA (analysis of variance) using SPSS statistical software (v.20). Each group was represented by 3 plants per pot (10 pots per group).

## 3. Results

### 3.1. Effect of Salinity Stress on Plant Survival Percentage

Growth and survival percentages of the Taif rose plant (*Rosa damascena trigintipetala Dieck*) decreased with increasing levels of NaCl stress, as compared with the control. Concerning the response of survival percentage due to NaCl treatments, the results clearly indicated that the highest survival percentage (98.3%) occurred in the control group (Table 1).

**Table 1.** Effect of due salinity stress on plant survival rate (% = 100 × survived plants/total plants).

| Parameters | NaCl Conc (x PPM) | | | | |
|---|---|---|---|---|---|
| | **0** | **250** | **500** | **1000** | **1500** |
| % Survived plants | 98.3 ± 0.55 [a] | 83.5 ± 0.85 [b] | 80.3 ± 0.49 [b] | 62.2 ± 0.78 [c] | 55.3 ± 2.17 [d] |

Data are expressed in group mean ± SE. Data with different letters show statistically significant ($p \leq 0.05$) difference. Least significant difference (LSD) = 1.62.

Culturing the Taif rose explants in pots containing 1500 ppm NaCl reduced the survival percentage to the minimum value (55.3%). The other treatments recorded a decrease in survival percentage with the increase in NaCl levels. However, the differences among all groups were statistically significant ($p \leq 0.05$), except between the 250 and 500 ppm groups. The value of the least significant difference was 1.62.

### 3.2. Effect of Salinity Stress on Plant Height

Table 2 shows that increased NaCl levels significantly ($p \leq 0.05$) reduced plant height (Table 2). Data also showed that 1500 ppm of NaCl gave the shortest shoots (38.9 ± 1.86 cm), whereas the control plants recorded the longest shoot length (72.5 ± 1.18 cm). Culturing the Taif rose explants at different concentrations of NaCl significantly ($p \leq 0.05$) decreased the plant height compared with control. The differences among group means were statistically significant, with 1.81 the least significant difference value.

**Table 2.** Effect of salinity stress on plant parameters (plant height, number of leaves, and number of flowers).

| NaCl Conc (x PPM) | Plant Parameters | | |
|---|---|---|---|
| | Plant Height | Number of Leaves | Number of Flower |
| 0 | 72.5 ± 1.18 [a] | 55.6 ± 0.67 [a] | 4.5 ± 0.32 [a] |
| 250 | 64.2 ± 1.16 [b] | 41.6 ± 0.48 [b] | 3.4 ± 0.15 [b,c,d] |
| 500 | 56.5 ± 0.35 [c] | 38.4 ± 0.12 [c] | 3.2 ± 0.07 [c] |
| 1000 | 47.7 ± 1.39 [d] | 28.5 ± 0.50 [d] | 3.6 ± 0.09 [d] |
| 1500 | 38.9 ± 1.86 [e] | 25.5 ± 0.31 [e] | 2.7 ± 0.12 [e] |
| LSD | 1.81 | 0.64 | 0.20 |

Data are expressed in group mean ± SE. Data in the same column annotated with different letters show statistically significant ($p \leq 0.05$) differences. Least significant difference (LSD).

### 3.3. Effect of Salinity Stress on Number of Leaves/Plant

Regarding the number of leaves (Table 2), the untreated Taif rose plants produced a significantly ($p \leq 0.05$) greater number of leaves per explant (55.6 ± 0.67) than with NaCl at all other concentrations. The averages of leaf numbers for salt concentrations (250, 500, 1000, and 1500 ppm) were 41.6, 38.4, 28.5, and 25.5 leaves per plant, respectively. All NaCl treatment decreased the number of leaves compared with the control. The differences between group means were statistically significant, with 0.64 the least significant difference value.

### 3.4. Effect of Salinity Stress on the Number of Flowers per Plant

Table 2 shows that plants grown at 1500 ppm NaCl significantly ($p \leq 0.05$) produced the least number of flowers (2.7 ± 0.12 rose/plant). While untreated plants produced the highest number (4.5 ± 0.32 rose/plant), it was clear that different NaCl concentrations were more effective on growth parameters. The minimum flower number (3.2 ± 0.07 rose/plant) was in the case of the 500 ppm NaCl dose. The differences among the group means were statistically significant ($p \leq 0.05$) except for the flower number (3.4, 3.2, and 3.6 rose/plant) at NaCl treatment doses of 250, 500, and 1000 ppm, respectively. The least significant difference between groups was 0.20 (Table 2).

### 3.5. Gas Chromatography Analysis of Essential Oil Contents in Taif Rose Plants

GC–MS analysis identified various compounds in Taif rose essential oils. The major components were identified and are listed in Figure 1. The total number of peaks were obtained from Taif rose plant leaves at A (0), B (250), and C (500) ppm NaCl, respectively, as illustrated in Figure 1.

Citronellol, geraniol, phenyl ethyl alcohol, nonadecene, and heneicosane were the major essential oil components of the Taif rose plant. Peaks in the different samples indicated that the plants grown under 500 ppm NaCl produced the highest contents of citronellol (16.56%), geraniol (8.67%) and phenyl ethyl alcohol (9.87%). The data in Table 3 also showed that there was an increase in the nonadecene content in untreated plants, which gave the highest value (8.13%) compared with the other NaCl levels. The heneicosane content increased with 250 ppm NaCl to the highest value and reached 13.12%, then decreased with increasing NaCl treatments and reached the lowest value (7.79%) with 1500 ppm NaCl. These results mean that the Taif essential oil levels were increased under some salinity stress treatments. These results explain the finding that NaCl at 500 ppm encouraged the cultured Taif rose plants to increase the citronellol, geraniol and phenyl ethyl alcohol contents (Table 3 and Figure 1).

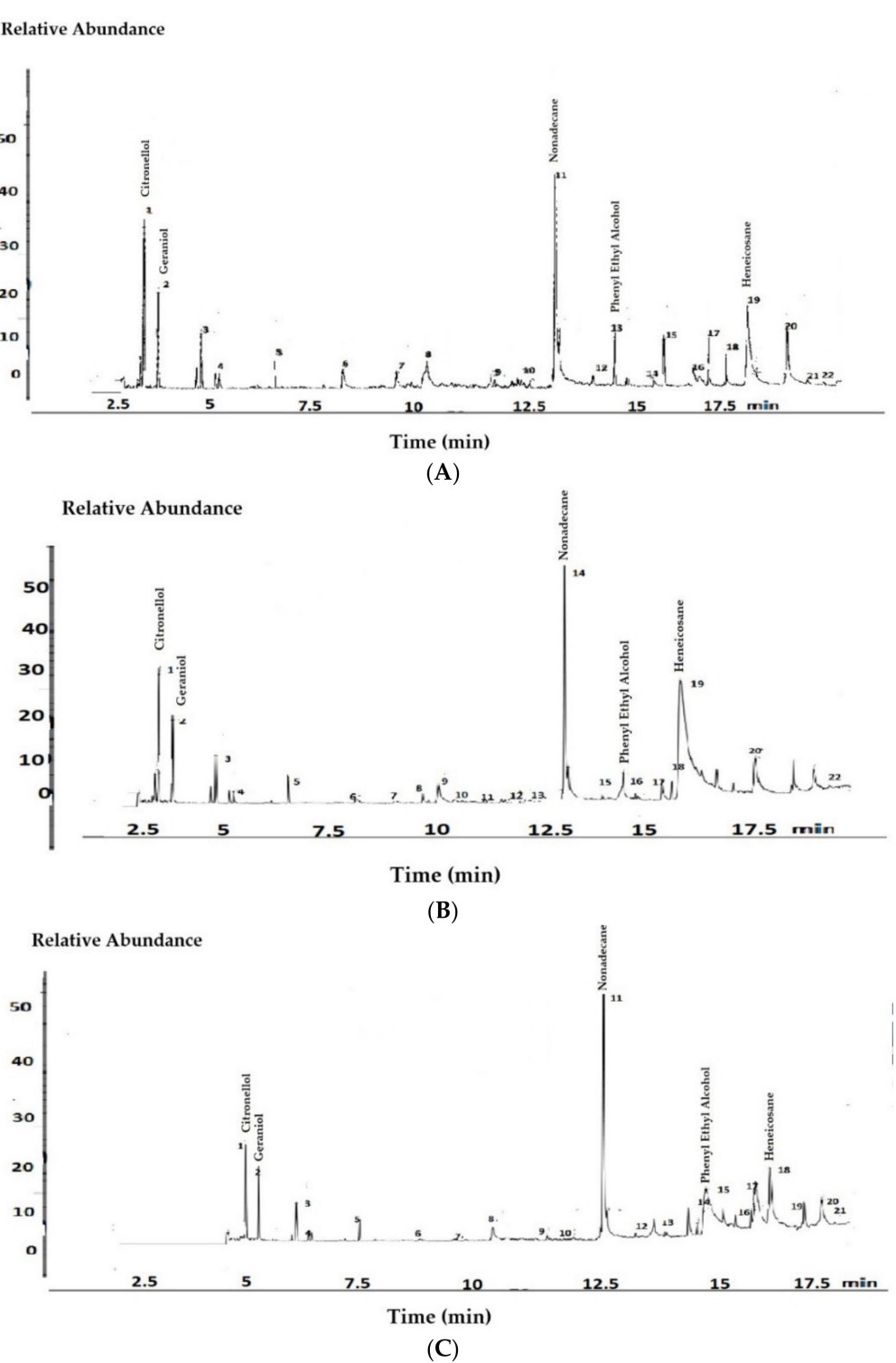

**Figure 1.** GC-MS chromatogram analysis of essential oil contents in Taif rose plants (*Rosa damascena trigintipetala Dieck*) grown under NaCl concentrations. (**A**) (0), (**B**) (250) and (**C**) (500) ppm NaCl.

**Table 3.** Effect of salinity stress on rose essential oil.

| NaCl Conc (x PPM) | Essential Oil (%) | | | | |
|---|---|---|---|---|---|
| | Citronellol | Geraniol | Phenyl Ethyl Alcohol | Nonadecane | Heneicosane |
| 0 | 14.77 ± 0.14 [a] | 7.87 ± 0.51 [a] | 5.15 ± 1.72 [a] | 8.13 ± 0.46 [a] | 9.22 ± 0.16 [a] |
| 250 | 10.26 ± 0.45 [b] | 4.94 ± 0.22 [b] | 5.15 ± 0.23 [a,b] | 6.32 ± 0.42 [b] | 13.12 ± 0.07 [b] |
| 500 | 16.56 ± 0.34 [c] | 8.67 ± 0.20 [a,c] | 9.87 ± 0.07 [c] | 3.23 ± 0.13 [c] | 8.71 ± 0.05 [a,c] |
| 1000 | 15.39 ± 0.41 [a,c,d] | 6.55 ± 0.32 [d,e] | 2.34 ± 0.19 [a,b,d] | 3.56 ± 0.31 [c,d] | 8.54 ± 0.31 [c,d] |
| 1500 | 14.98 ± 0.45 [a,d,e] | 7.72 ± 0.15 [a,c,e] | 7.43 ± 0.33 [b,c,e] | 3.51 ± 0.29 [c,d,e] | 7.79 ± 0.30 [e] |
| LSD | 0.53 | 0.44 | 1.12 | 0.48 | 0.3 |

Data are expressed as the mean of percentage at each group ± SE. Data in the same column annotated with different letters have statistically significant ($p \leq 0.05$) difference. Least significant difference (LSD).

## 4. Discussion

Roses are an important ornamental crop. The flower industry depends on flower color and fragrance [24]. The composition of the essential oils in flower extracts is an important determinant of oil quality [23].

The present study detected the effect of NaCl levels on the growth characteristics and essential oil content of *Rosa damascena Miller var. trigintipetala Dieck*, which is considered one of the best species for Taif rose oil production. This study confirmed that the growth characteristics of *Rosa damascena Miller var. trigintipetala Dieck* plants gradually increased in response to the elevation of NaCl levels. Meanwhile, a decrease in the number of leaves, survival percentage, and number of flowers as NaCl levels increased, compared with control plants, was in line with previous studies on the salinity of soil and irrigation water [4,25]. The results are also in good agreement with the results obtained by Sánchez-Montesinos et al., who demonstrated a reduction in plant growth of melon seedlings under saline stress conditions [26]. Plants under salt stress showed a decrease in shoot and root numbers, a delay in leaf appearance, and a reduction in leaf surface area and internode lengths [27].

Analysis of the oil samples using gas chromatography showed an increase in some essential oil contents in response to certain NaCl levels (500 ppm NaCl), and increased metabolic activities. Our results are in agreement with previous tests on Taif rose oil to find the main constituents, 31.3% citronellol, 19.5% geraniol, and 10.3% 5-methyl octadecane [11]. Others analyzed the components of damask rose essential oils and found oxygenated monoterpenes and aliphatic hydrocarbon as a major fraction. The major components were citronellol (35.5 to 49.2%), trans-geraniol (12.6 to 18.4%), nonadecane (8.9 to 15.1%), heneicosane (3.8 to 8.7%) and nonadecene (1.6 to 3.6%) [18,28]. However, the essential oils represented in the present study were geraniol, citronellol and citronellyl formate. Current data were also in agreement with reports that demonstrated the major chemical composition of rose geranium oils to be citronellol [29], and that aliphatic hydrocarbons, aliphatic esters, and alcohols are major volatiles in all rose genotypes [17,30]. Previous reports explained that the effects of salinity stress levels, as well as the degree of tolerance in rose flowers, could be due to the induction of the specific enzymes involved in the biosynthesis of these compounds by salinity [31].

## 5. Conclusions

Plant growth and the quality of the essential oil from *Rosa damascena Miller var. trigintipetala Dieck* were assessed at different NaCl levels. An obvious decrease response in plant growth was currently recorded. Moreover, this may represent an effective method of improving and stimulating the quality of essential oil constituents including citronellol, geraniol, and phenyl ethyl alcohol from *Rosa damascena Miller var. trigintipetala Dieck* at some NaCl levels, especially 500 ppm.

Chemical changes induced by salinity could reflect an adaptation response to this factor. In this case, exposing Taif rose plants to salinity stress may function as a potential technique that could increase essential oil production. An increase in Taif rose oil properties may be due to oil quality genetics rather than to environmental influences during plant growth. The current results could highlight the impact of salinity stress on *Rosa damascena Miller var. trigintipetala Dieck* to achieve better economic and industrial applications. It could also be applied to improving the yield of ornamental and other crops to be planted in the Taif region or other arid and semi-arid lands all over the kingdom for greater investment.

**Author Contributions:** Conceptualization, methodology, data curation, writing—original draft preparation, writing—review, editing: M.E.E.-S., M.M.M., and S.M.A.; funding acquisition: M.E.E.-S. All authors have read and agreed to the published version of the manuscript.

**Funding:** This research received no external funding.

**Data Availability Statement:** Not applicable.

**Acknowledgments:** This work was carried out using the facilities and materials at Taif University Researches Supporting Project number (TURSP-2020/139), Taif University, Taif, Saudi Arabia.

**Conflicts of Interest:** The authors declare no conflict of interest.

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
