# Peer review of "Oil and Flower Production in Rosa damascena trigintipetala Dieck under Salinity Stress in Taif Region, Saudi Arabia"

_sustainability, doi:10.3390/su13084547_

Round 1
Reviewer 1 Report
Introduction
General remarks
The authors should avoid the intensive use of paragraphs. For instance, in L 34-38 a paragraph made no sense because the first sentence in the next paragraph is the explanation of the last sentence of the paragraph before.
The authors should try to follow a “red thread” in their introduction, that structure their thoughts. At the moment it is more or less a selection of statements, although in general not wrong, that are just put together in a row.
The only part in the introduction that is clear, was the final sentence with the research goals. Unfortunately, it is not supported by the whole previous introduction. Therefore, the introduction need to undergo a thoroughly revision to support or justify the research goals.
Specific remarks
L25 The authors could specify respectively emphasize that salinity is the main limiting factor in arid areas with high evapotranspiration rates and/or due to intensive irrigation.
L26 Syntax is not correct, Sentence need to be revised.
L33 Syntax is not correct, Sentence need to be revised.
L55-56 This sentence doesn’t make sense
L60 What is the sense resp. the statement of this sentence?
Materials and methods
First paragraph need a thoroughly language revision. It is not readable in this state.
2.1 Experimental design
The experimental design is not sufficiently described!
How big are the pots, what volume/mass of substrate was filled in the pot, was the experiment carried out in a green house or in open field, what amount of water was applied, in which rates was the water applied and so on and so forth.
2.2 Measurements of growth parameters
It is not appropriate to say some morphological parameters are recorded. Please state what parameter, what units, was there only records at the end of the experiment or are the growth parameters are measured regularly?
2.3 Gas Chromatography of Taif rose Essential Oil
Extraction Method of essential oil, is there a specific method protocol that was used and described in literature. Is there a difference between essential oi and rose essential oil? Or why are there two different methods used? Again, please state if there is a specific protocol or method for extraction the essential oil.
2.4 Statistical analysis
It’s not clear what the triplicate means, from each pot one sample three times measured, from each pot three samples (flowers) taken and measured?
Which statistical methods are used? Also in the result section was not stated what method was used to calculate the significant differences between the treetments.
3 Results
Indeed, I found the results very interesting and informative. Nevertheless, they are not presented in an appropriate way.
The authors use Figure, fig. and Fig. (with a point without a point), in brackets or not in the text. This is not acceptable for a scientific manuscript.
The Tables and Figures should appear within the paragraphs they belong to, that would help to improve the readability of the text.
Table 1: The title is not fully self-explaining. What is the reference to the used percent? Flower dry weight, fresh weight or portion of oil types in total oil yield?
There are no numbers used to indicate significant differences but letters. And it is totally uncommon that significant different values have the same letter or numbers. This should be indicated by different letters.
All bars in all figures should be labeled with letters indicating if they are significant different or not only the not significant ones.
What is the meaning of least significant difference (LSD) this value was never introduced nor explained?
Figure 5 the y-axis is not labeled; the authors should align the three sub-figures so that at least the x-axis is adjusted to the same extend.
4 Discussion and 5 Conclusions
Before I am willing to consider a revision of the discussion and conclusion section, the previous thee section need to be thoroughly revised and improved by the authors. Otherwise, even if they have interesting results, it doesn’t make sense to invest any effort in revising these sections.
Author Response
Point 1: The authors should avoid the intensive use of paragraphs. For instance, in L 34-38 a paragraph made no sense because the first sentence in the next paragraph is the explanation of the last sentence of the paragraph before.
Response 1: The required modification has been completed
Point 2: The authors should try to follow a “red thread” in their introduction, that structure their thoughts. At the moment it is more or less a selection of statements, although in general not wrong, that are just put together in a row.
Response 2: Proofreading has been done
Point 3: The only part in the introduction that is clear, was the final sentence with the research goals. Unfortunately, it is not supported by the whole previous introduction. Therefore, the introduction need to undergo a thoroughly revision to support or justify the research goals.
Response 1: The required modification in introduction has been done
Specific remarks.
L25 The authors could specify respectively emphasize that salinity is the main limiting factor in arid areas with high evapotranspiration rates and/or due to intensive irrigation.
Response: The required modification has been done (L28-L30)
L26 Syntax is not correct, Sentence need to be revised.
Response: The required correct has been done
L33 Syntax is not correct, Sentence need to be revised.
Response: The required correct has been done
L55-56 This sentence doesn’t make sense
Response: The required modification has been done
L60 What is the sense resp. the statement of this sentence?
Response: The required modification in introduction has been done
Materials and methods
First paragraph need a thoroughly language revision. It is not readable in this state.
Response: The required correct has been done (L77)
2.1 Experimental design
The experimental design is not sufficiently described!
How big are the pots, what volume/mass of substrate was filled in the pot, was the experiment carried out in a green house or in open field, what amount of water was applied, in which rates was the water applied and so on and so forth.
Response: The required correct has been done(L83-L90)
2.2 Measurements of growth parameters
It is not appropriate to say some morphological parameters are recorded. Please state what parameter, what units, was there only records at the end of the experiment or are the growth parameters are measured regularly?
Response: The required correct has been done (L93-L95)
2.3 Gas Chromatography of Taif rose Essential Oil
Extraction Method of essential oil, is there a specific method protocol that was used and described in literature. Is there a difference between essential oi and rose essential oil? Or why are there two different methods used? Again, please state if there is a specific protocol or method for extraction the essential oil.
Response: The required correct has been done (L97)
2.4 Statistical analysis
It’s not clear what the triplicate means, from each pot one sample three times measured, from each pot three samples (flowers) taken and measured?
Which statistical methods are used? Also in the result section was not stated what method was used to calculate the significant differences between the treatments.
Response: The required correct has been done (L108-11)
3 Results
Indeed, I found the results very interesting and informative. Nevertheless, they are not presented in an appropriate way.
The authors use Figure, fig. and Fig. (with a point without a point), in brackets or not in the text. This is not acceptable for a scientific manuscript.
The Tables and Figures should appear within the paragraphs they belong to, that would help to improve the readability of the text.
Table 1: The title is not fully self-explaining. What is the reference to the used percent? Flower dry weight, fresh weight or portion of oil types in total oil yield?
There are no numbers used to indicate significant differences but letters. And it is totally uncommon that significant different values have the same letter or numbers. This should be indicated by different letters.
All bars in all figures should be labeled with letters indicating if they are significant different or not only the not significant ones.
What is the meaning of least significant difference (LSD) this value was never introduced nor explained?
Figure 5 the y-axis is not labeled; the authors should align the three sub-figures so that at least the x-axis is adjusted to the same extend.
Response: The required correct has been done
4 Discussion and 5 Conclusions
Before I am willing to consider a revision of the discussion and conclusion section, the previous three section need to be thoroughly revised and improved by the authors. Otherwise, even if they have interesting results, it doesn’t make sense to invest any effort in revising these sections.
Response: The required correct has been done (L190-L235)

Reviewer 2 Report
In the Abstract, the research methodology must be clarified in one sentence, for a better understanding.At Materials and Methods, Experiment design,the chemical description of the substrate used is missing. This can influence the results.
At Results all abbreviations in the tables must be explained in footnotes.
Author Response
Comments and Suggestions for Authors
Point 1: In the Abstract, the research methodology must be clarified in one sentence, for a better understanding.
Response 1: The required modification has been completed
Point 2: At Materials and Methods, Experiment design, the chemical description of the substrate used is missing. This can influence the results.
Response 2: The required modification has been completed
Point 3: At Results all abbreviations in the tables must be explained in footnotes.
Response 3 : The required modification has been completed

Reviewer 3 Report
No hypotheses stated. The experiment design is not applicable: no replications, no information about pot volume, plant and pot number per replication, growth conditions and time. Description of statistical analysis is not exhaustive. No bar meaning indication. Due to this circumstances results and statistical analysis presented in most figures seems controversial.
Author Response
Comments and Suggestions for Authors
Point 1: No hypotheses stated. The experiment design is not applicable: no replications, no information about pot volume, plant and pot number per replication, growth conditions and time.
Response 1: The required modification has been completed
Point 2: Description of statistical analysis is not exhaustive. No bar meaning indication. Due to this circumstances results and statistical analysis presented in most figures seems controversial.
Response 2: The required modification has been completed

Round 2
Reviewer 1 Report
The manuscript improved in the introduction and material & methods section substantially.
Nevertheless, there are still some mayor flaws in the manuscript the result section need still thoroughly improvements. At some points the authors did not implement the corrections required although they stated that they did.
Additonally the discussion and Conclusion section need a revision too.
All comments are shown in the attached document

Author Response
Response to Reviewer 1 Comments
Round 2
Point 1: Introduction: L33-34: In the Taif region or in arid regions, the authors might emphasize that. The authors did substantial improvements to the introduction section of the manuscript. Nevertheless, I did not get the clue how the paragraph (L50-60) contribute to the idea of the authors that salinity is one of the main factors driving oil composition, quantity or quality. 

Response 1: Change has been done (Line 12).
Point 2: Material & Methods: Again, the authors did improve the manuscript substantially.
I still have some questions.
2.1 Experimental design: Is there a specific reason why the plant five rose plant per pot? After the initial irrigation with 300 ml tap water, in the follow they irrigated each three days with water with different salt content. Was the amount of salt water also 300 ml per pot?
Response 2 (2.1): Corrections have been done (Line 87).
2.3 Extraction of essential oil: The reference [22] is only a secondary reference for this method, and the dichloromethane dissolution was not mentioned in this article. Could it be that there is a mistaking with reference [24],which suit better at this place, at least according to the title, since the article is not available to me.
Response 2 (2.3): Change has been done (Line 100-101).
Point 3: Results:
- There is still a mixture of different styles in addressing Tables and Figures for instance Figure, figure, Fig., Table and table. Please use Figure X and Table Y in captions as well as in the text, as stated in the authors guidelines of MDPI.
- Table 1 and Figure 1 are redundant, skip one. Table 2 and Figure 2 to 4 are redundant, skip either graphs or tables. If would suggest to keep the Tables, but correct the significance indicators. Otherwise, the missing information from table 2 need to be added to the graphs.
Response 3: Proofs have been done.
- The significance indicators in Table 1 and in the following Tables and Figures are not correct. First that are letters not numbers. Second five groups that are all significant different need also five different indicators. The stated indicators in the tables make no sense in this way.
- Changes have been done.
- Furthermore, the authors stated as least significant difference (LSD) = 1.62, so why is 250 and 500 not significantly different with a difference of at least 1.86?
- The LSD for survival rate was 1.62 (Line 122-123), however the difference between survival rates due to 250 and 500 equals 3.2 (it is not the least).
- Table 2, I cannot see why the number of flowers with 500 and 100 ppm NaCl are not significant different, with a LSD of 0.2 and a difference of at least 0.24.
- There is no treatment with 100ppm. In case you means the 1000ppm, there is a significant difference with the 500ppm dose, and the difference between means is 0.4.
- Please revise all significance indicators and significances thoroughly.
- It has been done.
Response 3 (3.1): Effect of salinity stress on plant survival%
L116-117 What does this sentence mean? 0 ppm NaCl is the control group and could therefore not be compared with itself? It shoul read “…the highest survival percentage (97.68%) occurred in the control group…”.
- proof has been done (Line118).
Response 3 (3.4): Effect of salinity stress on number of flowers /plants
L142-145 see comment above why are the numbers of flowers sig. different?
- proof has been done.
Response 3 (3.5): Gas Chromatography analysis of essential oil contents in Taif rose plant.
L148 there is no Figure 6. Figure 5 please include the salt content ppm in the graph, not only in the caption. Please align and adjust the 3 parts of the graph so that all axes are in one line and width.
- It is not clear what we could see in this graph the authors should indicate which peeks belong to each other, for example which are the peaks of specific oil contents (e.g. citronellol, geraniol or others).
- Therefore, it is inevitable to align the individual parts of the graph! This was already stipulated in the last review and, although stated by the authors, not done.
- Changes have been done.
Response 4: Discussion.
L200-203 The authors stated that other publications reported that salinity effects root growth, seed germination and so on. Since the authors did not include such parameters in their study, why are the now refer to them? This might be more interesting in the introduction section.
- In the result section the authors stated that some components of the rose oil changed with different salinity levels (L150-162). I am missing a discussion why this is the case or at least what could cause
- such changes. Additionally, is there evidence that quantity of oil production did change with different salinity stress level?
- The discussion is more a repetition of introduction and results than a discussion.
- Explanation has been added (Line 209-212).
Point 5: Conclusion:
- The authors stated that a specific level of salinity in the irrigation water could stimulate the quality of essential oil (L221-225). Here I missed in the discussion section and also in the result section a calculation or proof that (if there is one) the reduction of oil production because of decreased in rose biomass growth was compensated or even overcompensated by the amount or quality of the rose oil or its components.
- The current study focused on the weight and your criticism is very respectful to be considered in future work.
- I did not get the idea in the last sentence of the conclusions section, which should state and address the major outcome of the study. Here the authors mentioned ornamental application of salinity and other crops which are never mentioned before. The manuscript should close with a strong argument supporting or confuting the research hypotheses.
- Changes have been done.

Reviewer 2 Report
At Materials and Methods, Experiment design, the data on the chemical description of the used substrate (sand and peatmoss) is missing.
Author Response
Point: At Materials and Methods, Experiment design, the data on the chemical description of the used substrate (sand and peatmoss) is missing.
Response: Reference has been added (Line 85)
